
# Impact of atomic chlorine on the modelling of total methane and its $^{13}$C:$^{12}$C isotopic ratio at global scale

Joël Thanwerdas[1,*], Marielle Saunois[1], Antoine Berchet[1], Isabelle Pison[1], Didier Hauglustaine[1], Michel Ramonet[1], Cyril Crevoisier[2], Bianca Baier[3,4], Colm Sweeney[4], and Philippe Bousquet[1]

[1]Laboratoire des Sciences du Climat et de l'Environnement, CEA-CNRS-UVSQ, IPSL, Gif-sur-Yvette, France.
[2]Laboratoire de Météorologie Dynamique, École Polytechnique, IPSL, Palaiseau, France.
[3]Cooperative Institute for Research in Environmental Sciences (CIRES), University of Colorado-Boulder, Boulder, CO, USA 80305
[4]NOAA Earth System Research Laboratory Global Monitoring Division, Boulder, CO, USA 80305

**Correspondence:** J. Thanwerdas (joel.thanwerdas@lsce.ipsl.fr)

**Abstract.**

Methane (CH$_4$) is the second strongest anthropogenic greenhouse gas after carbon dioxide (CO$_2$) and is responsible for about 20% of the warming induced by long-lived greenhouse gases since pre-industrial times. Oxidation by the hydroxyl radical (OH) is the dominant atmospheric sink for methane, contributing to approximately 90% of the total methane loss. Chemical losses by

reaction with atomic oxygen (O$^1$D) and chlorine radicals (Cl) in the stratosphere are other sinks, contributing about 3% to the total methane destruction. Moreover, the reaction with Cl is very fractionating, thus it has a much larger impact on $\delta^{13}$C-CH$_4$ than the reaction with OH. In this paper, we assess the impact of atomic Cl on atmospheric methane mixing ratios, methane atmospheric loss and atmospheric $\delta^{13}$C-CH$_4$. The offline version of the Global Circulation Model (GCM) LMDz, coupled to a chemistry module including the major methane chemical reactions, is run to simulate CH$_4$ concentrations and $\delta^{13}$C-CH$_4$ at the

global scale. Atmospheric methane sink by Cl atoms in the stratosphere is found to be 7.32 ± 0.16 Tg/yr. Methane observations from vertical profiles obtained using AirCore samplers above 11 different locations across the globe and balloon measurements of $\delta^{13}$C-CH$_4$ and methane are used to assess the impact of the Cl sink in the chemistry transport model. Above 10 km, the presence of Cl in the model is found to have only a small impact on the vertical profile of total methane but a major influence on $\delta^{13}$C-CH$_4$ values, significantly improving the agreement between simulations and available observations. Stratospheric Cl is

also found to have a substantial impact on surface $\delta^{13}$C-CH$_4$ values, leading to a difference of + 0.27 ‰ (less negative values) after a 19-year run. As a result, this study suggests that the Cl sink needs to be properly taken into account (magnitude and trends) in order to better understand trends in the atmospheric $\delta^{13}$C-CH$_4$ signal when using atmospheric chemistry transport models for forward or inverse calculations.

## 1 Introduction

Methane (CH$_4$) is the second most powerful anthropogenic greenhouse gas after carbon dioxide (CO$_2$) and has a strong impact both on tropospheric and stratospheric chemistry. Its globally averaged concentrations have almost tripled since pre-industrial





times (Etheridge et al., 1998), exceeding 1850 ppb in 2017. The growth rate slowed in the 1990s and stabilized between 1999 and 2006. However, methane concentrations resumed increasing after 2006 and are still growing since then (Nisbet et al., 2019).

Due to large uncertainties in the estimation of methane sources and sinks (Saunois et al., 2016), the causes of this continued
increase in 2006 remain controversial (Saunois et al., 2017; Turner et al., 2019). Many studies used an inversion framework to optimize the sources and sinks in order to match atmospheric observations using the total methane information alone (Bergamaschi et al., 2018; Locatelli et al., 2015). However, total $CH_4$ mole fractions provide insufficient information to determine effectively the causes of this increase (Saunois et al., 2017). The $^{13}C:^{12}C$ isotopic methane ratio can be a powerful additional constraint to characterize a given methane source. For instance, analyzing the recent shift of this ratio to more negative val-
ues, (Nisbet et al., 2016, 2019) suggested that the methane increase since 2006 is driven by an increase in biogenic methane emissions. Using total $CH_4$ jointly with observations of $^{13}C:^{12}C$ ratio in inversions can potentially allow us to differentiate sources if their isotopic signature can be characterized well enough (Bousquet et al., 2006). This method has already been used in previous studies (Thompson et al., 2018; McNorton et al., 2018; Rice et al., 2016; Neef et al., 2010). However, those attempts of joint inversions do not reach exactly the same conclusions. Indeed, many parameters may influence the $^{13}C:^{12}C$
ratio such as the kinetic isotope effects associated with the sinks or the source isotopic signatures. The uncertainties and the regional variability of the isotopic signatures is an issue that should not be disregarded (Feinberg et al., 2018), especially for wetlands (Ganesan et al., 2018) that account for about 30 % of the total source and exhibit a distinctively light signature and a strong regional variability. In addition, due to the fractionation potential of chemical reactions in the atmosphere (McCarthy, 2003; Saueressig et al., 2001), the atmospheric isotopic ratio is largely affected by the intensity of the main sinks.

Three atmospheric species contribute to methane removal in the atmosphere : the hydroxyl radical (OH), electronically excited atomic oxygen atoms ($O^1D$) and the Cl radical (Cl). Oxidation by OH is the dominant sink and is responsible for about 90% of the total methane loss (Saunois et al., 2016). $CH_4 + O(^1D)$ and $CH_4 + Cl$ also contribute substantially to methane removal, especially in the stratosphere where $O(^1D)$ and Cl atoms are found in larger amounts than in the troposphere. Besides, the exceptionally large isotope fractionation in the reaction $CH_4 + Cl$ (1.066 compared with 1.0039 for $CH_4 + OH$ in this paper,
see Sect. 2.2) implies a significant effect on $^{13}C:^{12}C$ isotopic ratio values.

The impact of Cl (on methane as well as on $^{13}C:^{12}C$ isotopic ratio) has already been thoroughly studied in the troposphere and in the Marine Boundary Layer (MBL) (Wang et al., 2019; Hossaini et al., 2016; Allan et al., 2007; Allan, 2005; Wang et al., 2002; Allan et al., 2001). The Cl sink was found to account for 10% to more than 20% of total boundary layer methane oxidation and for 1% to 2.5% of total oxidation. However, Gromov et al. (2018) suggest that the effect of Cl is highly overestimated in
some tropospheric studies. Assessing an impact from Cl in the stratosphere was already attempted (Röckmann et al., 2004; McCarthy, 2003; McCarthy et al., 2001; Saueressig et al., 2001; Gupta et al., 1996; Müller et al., 1996) using box or 2-D models. Here we use a 3-D chemistry transport model to quantify methane loss through Cl oxidation, and the impact of atomic Cl in the stratosphere on global $CH_4$ concentrations, $XCH_4$ column and $^{13}C:^{12}C$ ratio using newly available 3-D Cl fields. A set of 115 vertical profiles of total methane retrieved during the 2010-2018 period using the AirCore technique (Karion et al.,





2010) at 11 different locations across the globe are used to assess potential improvements, especially in the stratosphere. The observations used for $^{13}C$:$^{12}C$ isotopic ratio comparison are balloon measurements presented in Röckmann et al. (2011).

Section 2 presents the model and data used to run the simulations as well as observations. In Section 3, we analyse first the impact of Cl on the total methane and then on methane isotopic ratio. Section 4 presents a discussion and conclusions.

## 2  Methods

### 2.1  The chemistry-transport model (CTM)

The LMDz Global Circulation Model (GCM) is the atmospheric component of the Institut Pierre-Simon Laplace Coupled Model (IPSL-CM) developed at the Laboratoire de Météorologie Dynamique (LMD) (Hourdin et al., 2006). The version of LMDz we use is an 'offline' version dedicated to the inversion framework created by Chevallier et al. (2005): precomputed meteorological fields provided by the online version of LMDz are given as inputs to the model, reducing significantly the computational time. The model is set up at a horizontal resolution of 3.8° x 1.9° (96 grid cells in longitude and latitude) with 39 hybrid sigma-pressure levels reaching an altitude up to about 75 km. About 20 levels are dedicated to the stratosphere and the mesosphere. The model time-step is 30 min and the output concentrations are 3-hourly averaged. The horizontal winds are nudged towards ECMWF meteorological analyses (ERA-Interim) in order to more realistically reproduce the actual meteorology. Vertical diffusion is parameterized by a local approach from Louis (1979), and deep convection processes are parameterized by the Tiedtke (1989) scheme.

The LMDz offline model, coupled with the SACS (Simplified Atmospheric Chemistry System) module (Pison et al., 2009), was previously used to simulate atmospheric concentrations of trace gas such as methane, carbon monoxide (CO), methyl-chloroform (MCF), formaldehyde ($CH_2O$) or dihydrogen ($H_2$) mixing ratios. For the purpose of this study, a new chemistry parsing system was developed (therefore replacing SACS) following the principle of the chemistry system in the regional model CHIMERE (Menut et al., 2013), allowing to provide user-specific governing system of chemistry reactions, thus generalizing the SACS module to any possible set of reactions. Each reaction is bound to a reaction type providing a way to compute the kinetic rate coefficients which depend on temperature and pressure. The algorithm updates the simulated species concentrations at each time step according to the reactions provided. The different species are either prescribed or simulated. The prescribed species are not transported in LMDz, nor their concentrations are updated depending on chemical production or destruction. They are only used to calculate reaction rates to update simulated species at each model time step. In this study, the isotopologues $^{12}CH_4$ and $^{13}CH_4$ of methane were simulated as separate tracers. Cl oxidation has also been implemented to complete the chemical removal of methane, which was accounted for only by OH + $CH_4$ and O($^1$D) + $CH_4$ in the SACS scheme. Here, $CH_4$ and its isotopologues are simulated, and OH, O($^1$D) and Cl distributions are prescribed from the INCA [INteraction with Chemistry and Aerosols] model (Hauglustaine et al., 2004) (see Sect. 2.3). In this study, we address here the sensitivity of $CH_4$ and its isotopologues to the presence of Cl through forward-modeling.



## 2.2 Reactions with OH, O($^1$D) and Cl and Kinetic Isotope Effect

Methane is removed from the atmosphere through chemical reactions with OH, O($^1$D) and Cl:

$$CH_4 + OH^\bullet \quad \rightarrow \quad CH_3^\bullet + H_2O \tag{R1}$$

$$CH_4 + Cl^\bullet \quad \rightarrow \quad CH_3^\bullet + HCl \tag{R2}$$

$$CH_4 + O(^1D)^\bullet \quad \rightarrow \quad CH_3^\bullet + OH^\bullet \tag{R3}$$

$$CH_4 + O(^1D)^\bullet \quad \rightarrow \quad H_2 + CH_2O \tag{R4}$$

OH is the main sink of methane, accounting for about 90% of total methane removal. The effect of Cl and O($^1$D) on total methane removal is significant only in the stratosphere. Due to the Kinetic Isotope Effect (KIE), the reaction rate coefficient associated with the reaction between methane and one of the three oxidants can vary from one isotope to another. The KIE is

defined by :

$$KIE = \frac{k_{12}}{k_{13}} \tag{1}$$

$k_{12}$ and $k_{13}$ denote the reaction rates for the reactions involving respectively $^{12}CH_4$ and $^{13}CH_4$. KIE values are usually greater than 1, meaning that the oxidant reacts faster with the lighter isotope. The reaction rates are taken from Burkholder et al. (2015) and the KIEs from Saueressig et al. (1995) for the reaction with Cl and from Saueressig et al. (2001) for OH and O($^1$D). All the

reactions and associated values are reported in the Table 1. Few studies have focused on assessing the KIE for $CH_4$ chemical sinks (especially for O($^1$D) and Cl) within a wide temperature range and large uncertainties still remain. McCarthy (2003) suggests to use those values for stratospheric methane simulations.

**Table 1.** Rate constants and KIEs of the methane sinks.

| Reactions | KIE | Reference | Reaction constant | Reference |
|---|---|---|---|---|
| OH | 1.0039 | Saueressig et al. (2001) | $2.45 \times 10^{-12} \cdot$ exp(-1775/T) | Burkholder et al. (2015) |
| O($^1$D) - R3 | 1.013 | Saueressig et al. (2001) | $1.125 \times 10^{-10}$ | Burkholder et al. (2015) |
| O($^1$D) - R4 | 1.013 | Saueressig et al. (2001) | $3.75 \times 10^{-11}$ | Burkholder et al. (2015) |
| Cl | $1.043 \cdot$ exp(6.455/T) | Saueressig et al. (1995) | $7.1 \times 10^{-12} \cdot$ exp(-1280/T) | Burkholder et al. (2015) |

Besides chemical removals, $CH_4$ soil uptake sink is accounted in this study as a negative source and its KIE is used to define an effective isotopic signature (see Sect. 2.5).

## 2.3 Atomic Cl field

Tropospheric and stratospheric Cl have different origins. Tropospheric Cl originates mostly from sea salt, organochlorines and open fires (Wang et al., 2019; Hossaini et al., 2016). The most important inorganic Cl compound in the troposphere is hydrogen



chlorine (HCl). Its principal source is acid displacement from sea salt aerosol. HCl is quickly removed from the troposphere by wet deposition due to its high solubility in water. Hence, it is not the main source of stratospheric Cl. The majority of the Cl found above the tropopause is released from long-lived Cl containing species. Man-made organochlorines such as chlorofluorocarbons (CFCs) and hydrochlorofluorocarbons (HCFCs) (Von Clarmann, 2013; Nassar et al., 2006) along with

methyl chloroform ($CH_3CCl_3$) and carbon tetrachloride ($CCl_4$) represent about 80 % of the source of Cl in the stratosphere. The rest (15-20 %) comes from methylchloride ($CH_3Cl$) which is mostly natural. The most abundant CFCs and HCFCs found in the stratosphere are currently CFC-11, CFC-12, CFC-113 and HCFC-22. These compounds release reactive Cl by photolysis or oxidation in the stratosphere and mesosphere. The two main Cl reservoir species in the stratosphere are HCl and chlorine nitrate ($ClONO_2$). Atomic Cl is converted to HCl mainly through the reaction with $CH_4$ (R2).

Since the 1987 Montreal Protocol, a decline in the emissions of anthropogenic long-lived Cl-containing species and consequently of the HCl concentrations in the stratosphere have been observed [Bernath and Fernando, 2018].

     We use the LMDz-INCA model to simulate beforehand the OH, O($^1$D) and Cl three-dimensional and time dependant concentrations to be prescribed in our methane simulations. LMDz-INCA couples the INCA [INteraction with Chemistry and Aerosols] (Folberth et al., 2006; Hauglustaine et al., 2004) and the LMDz GCM described in Section 2.1. Seventeen ozone-

depleting substances made up of five CFCs (CFC-12, CFC-11, CFC-113), three HCFCs (HCFC-22, HCFC-141b, HCFC-142b), two halons (Halon-1211, Halon-1301), $CH_3CCl_3$, $CCl_4$, $CH_3Cl$, methylene chloride ($CH_2Cl_2$), chloroform ($CHCl_3$), methyl bromide ($CH_3Br$) and HFC-134a and the associated photochemical reactions are included in the INCA chemical scheme to produce Cl (Terrenoire et al., in preparation, 2019). In the LMDz-INCA simulations, the surface concentrations of these long-lived Cl source species are prescribed according to the historical datasets prepared by Meinshausen et al. (2017). The final Cl

field will be referred to hereinafter as LMDz-INCA field.

     Figure 1 shows the key features of the resulting atomic Cl climatological fields averaged for the 2003-2009 period. We use a climatological field for Cl due to a lack of simulations for more recent years and to avoid uncertainties in simulated interannual variabilities and trends in Cl concentrations. However a sensitivity test was performed to assess whether the recent decline of Cl could have a substantial impact on $^{13}$C:$^{12}$C ratio values. This is discussed further in Section 3.5.2. The global

mean volume-weighted stratospheric Cl atom concentration is about $3.25 \times 10^5$ atoms.cm$^{-3}$. Acid displacement from sea salt aerosol (main source of tropospheric Cl) was not implemented in the model because the study mainly focuses on stratospheric Cl. Therefore, the tropospheric background mean (240 atoms.cm$^{-3}$) is smaller than the results of studies focusing specifically on troposphere such as Hossaini et al. (2016) which presents a background of 1300 atoms.cm$^{-3}$ or Wang et al. (2019) with a background of 620 atoms.cm$^{-3}$. Nevertheless, Hossaini et al. (2016) concentrations well above 1000 atoms.cm$^{-3}$ at the surface

might be overestimated according to Gromov et al. (2018). Figure 1b shows the time evolution of the atomic Cl concentrations over the climatological year. The processes by which increased photolysis within polar vortices leads to more active Cl are well-represented and should be a key parameter of the seasonal variability of simulated stratospheric methane in those regions. Lower latitude regions do not present such large fluctuations.





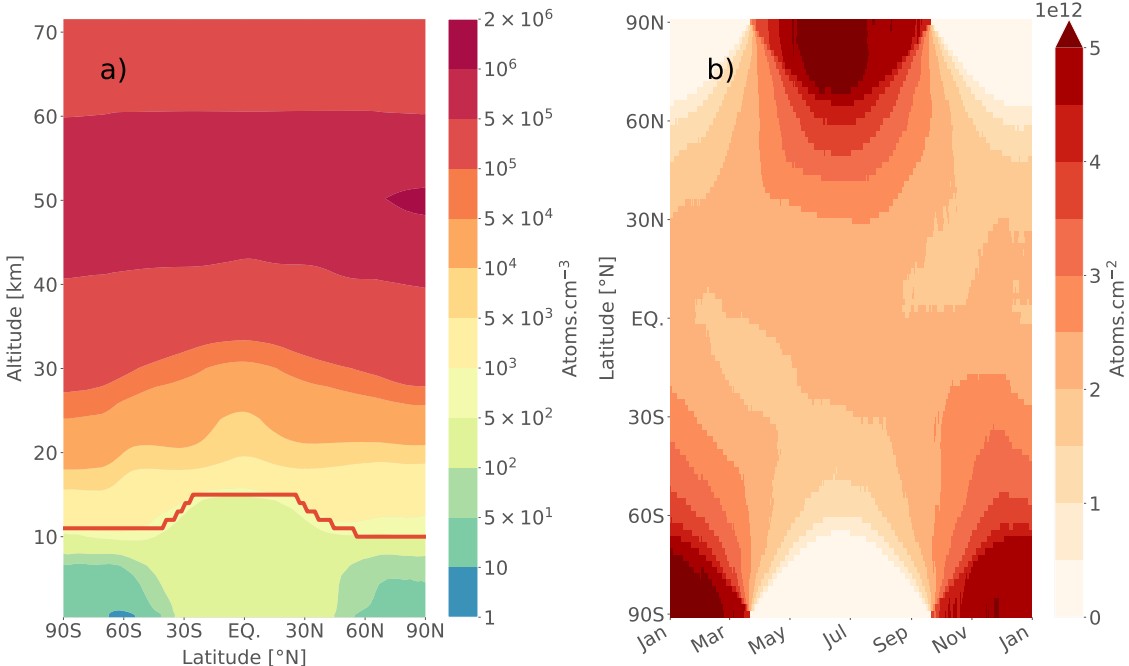

**Figure 1.** 3-D LMDZ-INCA climatological field of Cl for the 2003-2009 period. a) Meridional cross-section of Cl concentrations. The red line represents the mean tropopause level. b) Time evolution of meridional Cl total column. The total column have been computed using volume-weighted integration.

## 2.4 Total CH$_4$ surface fluxes

The surface fluxes prescribed to simulate total methane mixing ratios are the ones suggested by the Global Carbon Project (GCP) as a priori emissions for inversions performed for the Global Methane Budget (Saunois et al., 2019). Anthropogenic (including biofuels) and fire emissions for the 2000-2017 period are taken from bottom-up estimates provided by the EDGARv432

5   (http://edgar.jrc.ec.europa.eu/overview.php?v=432SECURE=123, Janssens-Maenhout et al., 2017) and the GFED4s databases (van der Werf et al., 2017), respectively. Statistics from British Petroleum (BP) and the Food and Agriculture Organization of the United Nations (FAO) have been used to extend the EDGARv432 database, ending 2012, until 2017. The natural sources emissions are based on averaged literature values : Poulter et al. (2017) for the wetlands, Ridgwell et al. (1999) for the soil sink, Kirschke et al. (2013) for the termites, Lambert and Schmidt (1993) for the ocean and Etiope (2015) for geological sources.

10   All the sectors and their emission intensities averaged over the 2006-2018 period are listed in Table 2. The emissions for 2018 have been set equal to 2017. Figure S2 shows the evolution over time of methane emissions for each sector. CH$_4$ emissions from natural sources are kept constant over the period of time considered while anthropogenic emissions linearly increase. The anthropogenic sources account for 61 ± 2 % (1$\sigma$ for interannual variability) and the natural sources for 39 ± 2 % of the total global budget.



## 2.5   $\delta^{13}$C signatures

The $^{13}$C:$^{12}$C ratio is reported through the $\delta^{13}$C value defined by :

$$\delta^{13}C_{\text{sample}} = \frac{R}{R_{\text{std}}} - 1 \tag{2}$$

$R$ is the $^{13}$C:$^{12}$C ratio of the sample and $R_{\text{std}}$ is the standard $^{13}$C:$^{12}$C ratio taken as a reference scale. In this paper, the signature is computed according to the Vienna - Pee Dee Belemnite (V-PDB) standard reference ratio ($R_{V-PDB}$ = 0.0112372 mol.mol$^{-1}$) (Craig, 1957). The two species $^{12}$CH$_4$ and $^{13}$CH$_4$ are simulated separately in LMDz. Each individual source is assigned an isotopic signature and the amount of $^{12}$CH$_4$ and $^{13}$CH$_4$ emitted from this source can be easily inferred from its signature and the intensity of the total methane emission from this source using the equations below :

$$F_{13} = (1 + \delta^{13}C)R_{\text{std}} \times \frac{M_{13}}{M_{12}} \times F_{12} \tag{3}$$

$$\frac{F_{12}}{M_{12}} + \frac{F_{13}}{M_{13}} = \frac{F_{\text{TOT}}}{M_{\text{TOT}}} \tag{4}$$

$F_{\text{TOT}}$, $F_{13}$ and $F_{12}$, denote the total CH$_4$, $^{13}$CH$_4$ and $^{12}$CH$_4$ mass fluxes, respectively. $M_{\text{TOT}}$, $M_{13}$ and $M_{12}$ correspond to their molar masses. Unlike the other sinks, we consider the soil sink as a negative source at the surface. Therefore we define an effective $\delta^{13}$C-CH$_4$ signature ($\delta^{13}C_{eff}$) based on the KIE of the soil sink and the mean $^{13}$C:$^{12}$C ratio at surface :

$$\delta^{13}C_{eff} = \frac{1 + \delta^{13}C_{\text{amb}}}{\text{KIE}_{\text{soil}}} - 1 \tag{5}$$

$\delta^{13}C_{\text{amb}}$ denotes the atmospheric isotopic signal near the surface. A mean value of − 47.2 ‰ set constant over time is prescribed. This value is in good agreement with the observed records (see Sect. 3.5). A KIE$_{\text{soil}}$ of 1.020 (Snover and Quay, 2000; Reeburgh et al., 1997; Tyler et al., 1994; King et al., 1989) was chosen. It leads to a $\delta^{13}C_{eff}$ of -65,9 ‰. The signatures of each sector are based on literature average values and summarized in Table 2. We use region-specific signatures over the 11 regions of the TransCom project (see Figure S4) for wetlands, gas and biofuels-biomass sectors that exhibit a strong regional variability of signatures. The signature is set globally uniform for the other sectors. More information about the regional isotopic signature and the references are provided in the supporting information (Text S1). The isotopic signatures of wetland emissions are inferred from multiple regional studies around the world and aggregated into a 11-regions map, to be consistent with the other sectors. Initially, this wetlands isotopic signature map was used to run the first simulations but we realized that the flux- and area-weighted signature can appear high compared to other papers (Feinberg et al., 2018; Ganesan et al., 2018) which studied isotopic signature values at a finer resolution. Overestimated isotopic signatures in boreal regions may be the cause of this difference. Therefore, the isotopic signature map from Ganesan et al. (2018) has also been used to assess the impact of a lower global isotopic signature. In Sect. 3.5.1, it is briefly shown that taking this lower signature does not affect the conclusions of our paper.





**Table 2.** Intensities and mean (flux- and area-weighted) isotopic signatures of methane sources averaged over 2000-2018. The sources with a * symbol are prescribed with a regional variability. The value given here is the mean over the 11 regions.

| Natural sources and sinks | Intensity (Tg/yr) | $\delta^{13}$C (‰) | Anthropogenic sources | Intensity (Tg/yr) | $\delta^{13}$C (‰) |
|---|---|---|---|---|---|
| Wetlands | 180.2 | -58.6* | Livestock | 112.3 | -61 |
| Termites | 8.7 | -63 | Rice cultivation | 36.0 | -63 |
| Ocean | 14.4 | -42 | Oil, Gas, Industry | 71.2 | -39.7* |
| Geological (onshore) | 15.0 | -50 | Biofuels - Biomass | 27.6 | -25.8* |
| Soil | -37.9 | -65.9 | Waste | 66.1 | -49.7 |
| | | | Coal | 32.5 | -35 |

## 2.6 Observations

### 2.6.1 AirCore measurements

An original set of 115 total methane vertical profiles retrieved above 11 different locations during the 2012-2018 period is used to compare the results of the simulations to observed values. 80 profiles (NOAA GGGRN AirCore_v20181101 dataset) are provided by the NOAA-ESRL Aircraft Program (Karion et al., 2010) and 35 by the French AirCore Program (Membrive et al., 2017). Those vertical profiles have all been collected using the AirCore technique (Karion et al., 2010). This technique retrieves air samples going from the surface to approximately 30km. The locations and the number of profiles associated are shown in Figure 2. This dataset has also been used in this study to analyze the seasonal vertical trend of $CH_4$ in both the troposphere and stratosphere and the model ability to reproduce the observations depending on the season. A summary of the information about the provider, the location and the number of profiles retrieved at this location is given in Table 3. The CRDS analyzer precision of the measurements of the AirCore sample for both NOAA and French AirCore Program is less than 2 ppb for $CH_4$ which is generally much smaller than the model-observation mismatch.

### 2.6.2 Balloon vertical profiles of $\delta^{13}$C

We use air samples from stratospheric balloon flights from Röckmann et al. (2011) to compare the simulated $\delta^{13}$C-$CH_4$ to observations. The information is summarized in Table 4. The samples were retrieved at four different locations going from subtropical to high latitudes, above an altitude of 10 km and up to 35 km. We assume that $\delta^{13}$C-$CH_4$ vertical profiles do not exhibit a strong inter-annual variability and therefore we can compare our simulations and the observations retrieved before 2000.

### 2.6.3 Surface $\delta^{13}$C observations

Multiple surface stations from the Global Greenhouse Gas Reference Network (GGGRN), part of NOAA-ESRL Global Monitoring Division (NOAA-ESRL GMD), are collecting air samples on an approximately weekly basis. Those air samples are





**Table 3.** Providers, locations and number of vertical profiles of total methane retrieved using AirCore technique between 2012 and 2018.

| Provider | Location | Number of profiles | Longitude | Latitude |
|---|---|---|---|---|
| NOAA-ESRL Aircraft Program | Edwards AFB/Dryden, USA | 6 | -117 | 34 |
| | Boulder, CO, USA | 33 | -103 | 40 |
| | Lamont, OK, USA | 30 | -97 | 36 |
| | Park Falls, WI, USA | 4 | -90 | 46 |
| | Sodankyla, Finland | 6 | 26 | 67 |
| | Lauder, NZ | 1 | 169 | -45 |
| French AirCore Program | Alice Springs, Australia | 3 | 133 | -23 |
| | Aire-sur-l'Adour, France | 9 | 1 | 43 |
| | Trainou, France | 17 | 2 | 48 |
| | Timmins, Ontario, Canada | 4 | -83 | 48 |
| | Esrange, Northern Sweden | 2 | 21 | 67 |

**Table 4.** Overview of balloon flights and number of samples analyzed for $\delta^{13}$C. Each flight is given a flight ID as STA-JJ-MM, where STA is the 3-letter-code for the balloon launch station, JJ the year and MM the month of sampling. This Table is adapted from Röckmann et al. (2011). [1] HYD: Hyderabad, India (17.5 °N, 78.60 °E); [2] KIR: Kiruna, Sweden (67.9 °N, 21.10 °E); [3] ASA: Aire sur l'Adour, France (43.70 °N, 0.30 °E); [4] GAP: Gap, France (44.44 °N, 6.14 °E);

| Flight ID | Flight Date | Location | $^{13}$C | Characteristics |
|---|---|---|---|---|
| Flights operated by MPI für Sonnensystemforschung | | | | |
| HYD-87-03 | 03/26/87 | HYD[1] | 5 | Subtropical |
| KIR-92-01 | 01/18/92 | KIR[2] | 13 | Artic weak vortex, final warming series |
| KIR-92-02 | 02/06/92 | KIR[2] | 10 | Final warming series |
| KIR-92-03 | 03/20/92 | KIR[2] | 10 | Final warming series |
| ASA-93-09 | 09/30/93 | ASA[3] | 15 | Mid-latitudinal background |
| KIR-95-03 | 03/07/95 | KIR[2] | 15 | Artic with mid-latitudinal characteristics |
| HYD-99-04 | 04/29/99 | HYD[1] | 10 | Subtropical |
| GAP-99-06 | 06/23/99 | GAP[4] | 15 | Mid-latitudinal summer |
| Flights operated by Institut für Meteorologie und Geophysik, Universität Frankfurt | | | | |
| KIR-00-01 | 01/03/00 | KIR[2] | 13 | Artic strong vortex |
| ASA-01-10 | 10/11/01 | ASA[3] | 13 | Mid-latitudinal background |
| ASA-02-09 | 09/15/02 | ASA[3] | 13 | Mid-latitudinal background |
| KIR-03-03 | 03/06/03 | KIR[2] | 13 | Arctic vortex, mesospheric enclosure |
| KIR-03-03 | 06/09/03 | KIR[2] | 13 | Artic summer |



analyzed by the Institute of Arctic and Alpine Research (INSTAAR) to provide isotopic measurements of $CH_4$. 18 stations that recorded enough values during the 2000-2018 period to estimate a $\delta^{13}C$-$CH_4$ range over this period and the negative trend that started around 2007 were selected. Only monthly values were aggregated. The locations of the selected stations are given in Figure 2 (green circles). These observations will be referred to as NOAA-GGGRN surface observations.

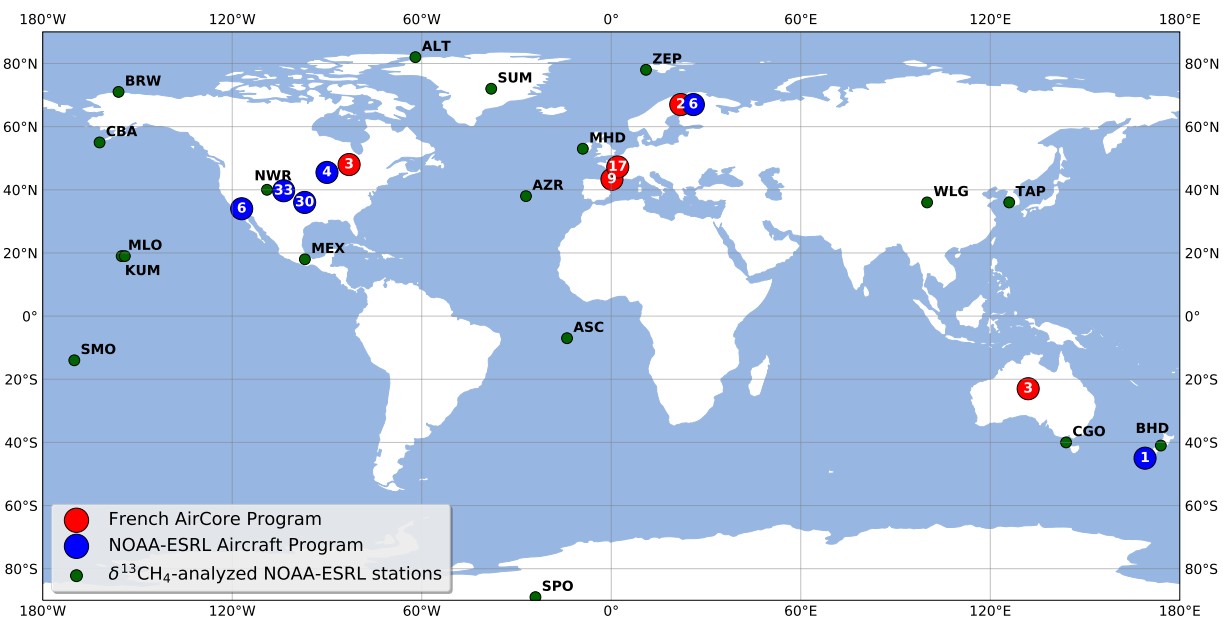

**Figure 2.** Locations of AirCore vertical profiles retrievals and NOAA-GGGRN stations. The number inside each AirCore location markers indicates the number of different profiles retrieved at each location.

## 2.7 Simulations performed

### 2.7.1 Total methane simulations

To assess the impact of Cl in the stratosphere on total methane, we perform two simulations with TOT_CHL and without TOT_REF the Cl sink implemented (Table 5). We simulate total methane mixing ratios for the 2006-2018 period starting from the same initial conditions, and use the varying GCP emissions (see Sect. 2.5). The initial conditions have been obtained using optimized fluxes from results of inversions performed by Locatelli et al. (2015). The simulated vertical profiles of total methane are compared to AirCore measurements in Sect. 3.2.

### 2.7.2 Methane isotope simulations

We simulated in parallel $^{12}CH_4$ and $^{13}CH_4$ concentrations, so that the sum of $^{12}CH_4$ and $^{13}CH_4$ is set equal to total $CH_4$. Since $\delta^{13}C$-$CH_4$ simulated values need a larger time to adjust (Tans, 1997), we first performed a 19-year spin-up using constant



emissions of year 2000 including all chemical sinks and starting in 2000 in order to obtain a good spatial and vertical distribution of $\delta^{13}$C-CH$_4$ values . Then we adjusted the simulated isotopic signal to the available NOAA-GGGRN $\delta^{13}$C-CH$_4$ surface observations (see Sect. 2.6.3) in order to obtain satisfying initial conditions. From these initial conditions, we run an ensemble of scenarios for 19 years (2000-2018) using constant sources of year 2000, and varying the chemical sink of methane through
Cl. The characteristics of each sensitivity test are summarized in Table 5.

| Simulation name | Traced species | Cl-CH$_4$ Reaction | Simulation period | Tropospheric chlorine | MBL chlorine | Stratospheric chlorine | Decreasing chlorine |
|---|---|---|---|---|---|---|---|
| TOT_CHL | total CH$_4$ | YES | 2006-2018 | 240 cm$^{-3}$ | 130 cm$^{-3}$ | YES | NO |
| TOT_REF | total CH$_4$ | NO | 2006-2018 | 240 cm$^{-3}$ | 130 cm$^{-3}$ | YES | NO |
| d13_CHL | $^{12}$CH$_4$ - $^{13}$CH$_4$ | YES | 2000-2018 | 240 cm$^{-3}$ | 130 cm$^{-3}$ | YES | NO |
| d13_REF | $^{12}$CH$_4$ - $^{13}$CH$_4$ | NO | 2000-2018 | 240 cm$^{-3}$ | 130 cm$^{-3}$ | YES | NO |
| **Sensitivity tests** | | | | | | | |
| S5 | $^{12}$CH$_4$ - $^{13}$CH$_4$ | YES | 2000-2018 | NO | NO | YES | NO |
| S6 | $^{12}$CH$_4$ - $^{13}$CH$_4$ | YES | 2000-2018 | 240 cm$^{-3}$ | 130 cm$^{-3}$ | NO | NO |
| S7 | $^{12}$CH$_4$ - $^{13}$CH$_4$ | YES | 2000-2018 | 620 cm$^{-3}$ | 335 cm$^{-3}$ | YES | NO |
| S8 | $^{12}$CH$_4$ - $^{13}$CH$_4$ | YES | 2000-2018 | 620 cm$^{-3}$ | 1200 cm$^{-3}$ | YES | NO |
| S9 | $^{12}$CH$_4$ - $^{13}$CH$_4$ | YES | 2000-2018 | 240 cm$^{-3}$ | 130 cm$^{-3}$ | YES | 5%/decade |
| S10 | $^{12}$CH$_4$ - $^{13}$CH$_4$ | YES | 2000-2018 | 620 cm$^{-3}$ | 1200 cm$^{-3}$ | YES | 5%/decade |

**Table 5.** Nomenclature and characteristics of the simulations. See Sect. 3.5 for more information about the sensitivity tests.

# 3   Results

## 3.1   Impact of Cl sink on total methane mixing ratios

We analyze the simulated methane mixing ratios of TOT_REF and TOT_CHL during the 2006-2018 period. The total stratospheric sink for methane is found to be equal to 24.68 ± 0.73 (1$\sigma$ for interannual variability) Tg/yr on average over the
2006-2018 period of simulation ($\sim$ 5.3 % of total atmospheric sink). The simulations yield a stratospheric Cl sink of 7.16 ± 0.27 Tg/yr, thus accounting for about 29% of the total stratospheric sink. This result is at the lower end of the range estimated by former papers for the period 2000-2009 (Kirschke et al., 2013) [Saunois et al. 2019] : [16-84] Tg for total stratospheric loss with a Cl contribution between 20 and 35%.

Figure 3 shows the relative contribution of each sink to the total sink in each latitude/altitude box of the model. We use
the lapse rate (2 K/km) definition from the World Meteorological Organization (WMO) to define the tropopause. Below the tropopause, OH is mostly responsible for the removal of methane (97% of the chemical sink) as expected, but it contributes only for 45 % on average in the stratosphere : the rest of the methane sink is due to reactions with Cl and O($^1$D). We estimate that the

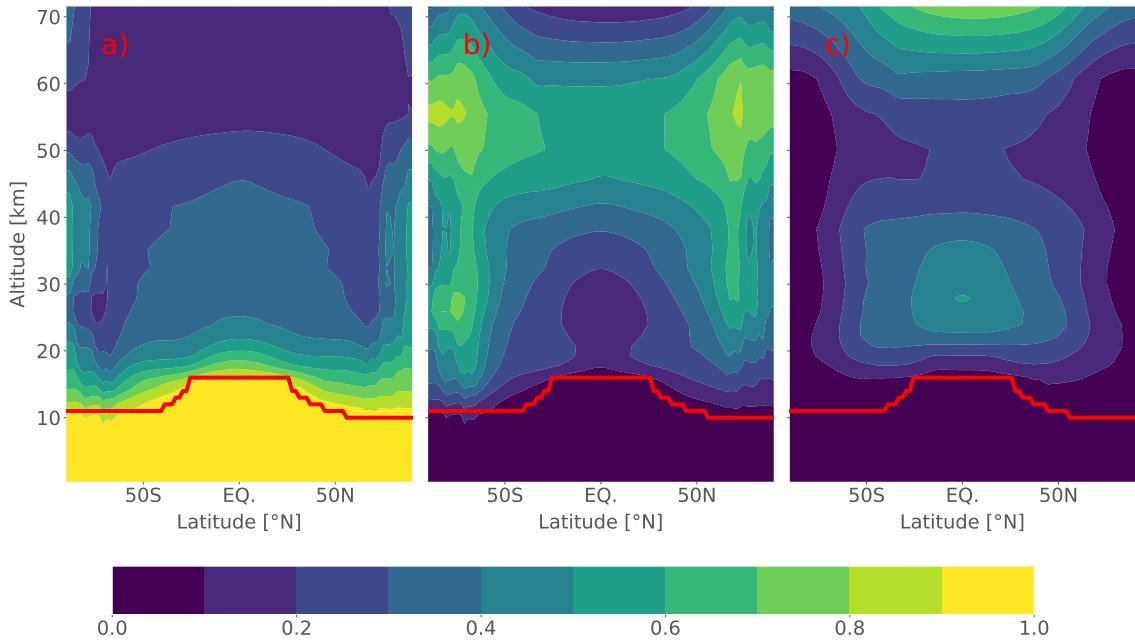

**Figure 3.** Global distribution of the relative contribution of each methane sink to the total chemical sink. The green line indicates the mean tropopause level. a) OH contribution to the sink. b) Cl contribution. c) O($^1$D) contribution.

Cl sink can reach 40-60 % of the total sink strength at high latitudes between 20 and 40 km of altitude and 60-80% above 40 km. Between 20 and 40 km altitude, the sink associated with O($^1$D) is responsible for half of the total sink in tropical regions due to the high $O_3$ photolysis rates there but only 30-40% at higher altitudes where enhanced Cl concentrations contribute 50-60% to the total chemical sink. Note that, as explained in Section 2.3, our Cl concentrations in the troposphere are likely to

5   be underestimated and its relative overall contribution of 1.23 ± 0.02 Tg/yr as well.

The change in methane mixing ratio due to the implementation of the Cl sink does not exceed 10 ppb in the troposphere (0,5 % of the tropospheric mixing ratio). However, it can lead to a reduction of up to 225 ppb between 50 and 60 km ($\sim$ 50% of the mixing ratio at this level) as well as 100 ppb above 20 km in the extratropics and 30 km in the tropics ($\sim$ 8 %).

These results confirm that the Cl sink in the stratosphere accounts for a large proportion of the simulated methane chemical

10  sink in some regions of the atmosphere.

### 3.2   Impact of Cl on methane total column

The impact of Cl on the dry-air column average mole fraction of methane (XCH$_4$) is also an important element to consider. This quantity is computed as the ratio of methane column density to dry-air column density. The value is here determined using a dry-air weighted mass average of the mixing ratio. Dry-air column average mole fraction will be referred to hereinafter as





the simple term 'column'. Remotely sensed total column retrievals are often used in inversions as additional constraints (or as an independent validation) and errors in the total column modeling due to the missing Cl sink could significantly influence the results of an inversion.

We assess that after a 12-year simulation from 2006 to 2018, the difference of global mean total columns between TOT_REF

and TOT_CHL is 18 ± 0.4 ppb (1$\sigma$ for spatial and time variability) for the year 2018 which represents a 1% difference. This difference tends to increase by about 1 ppb/yr over the 2012-2018 period and 1.5 ppb/yr over the 2006-2012 period. The difference is due to an adjustment time and a difference of source intensities between the two periods. Figure 4 shows the total column regional variability averaged over the year 2018 and the differences between TOT_REF and TOT_CHL. It also includes the tropospheric and stratospheric partial columns differences (Fig. 4c and Fig. 4d). The difference in tropospheric columns

is as high as 10.6 ppb between the two simulations, but with only a very small geographical variability of the tropospheric column difference (of typically 0.1 ppb around 10.6 ppb, see Figure 4c). Most of the total column regional variability comes from the stratosphere. The stratospheric partial column difference has a zonal distribution. It shows a variability of 7.7 ppb (1$\sigma$ for spatial variability) and the maximum values (up to 60 ppb) are found in high-latitude regions. The extreme values found over Greenland and Antarctica are strongly correlated to the orography. The ratio of the mass in the stratosphere to that in the

total column is larger in regions where the surface is higher and, in addition, the Cl sink has larger impact in the stratosphere. Therefore, the total methane column difference shows greater values (difference of more than 5 ppb) over those high regions. However, they do not strongly influence the mean column averaged over the globe since the total dry-air mass above is smaller.

The total column difference (Fig. 4b) shows difference reaching 1% to 2% of the methane column after a 12-year simulation. This result may change depending on the averaging kernel used for comparison to vertical profiles retrieved by remote sensing

techniques.

### 3.3 Total methane vertical profiles

The set of total methane vertical profiles retrieved using the AirCore technique is used to assess how much the implementation of the Cl sink in the model improves simulations. Since the methane mixing ratio reduction due to the Cl sink is proportional to the amount of methane present in the atmosphere, the reduction ratio is more relevant to assess the impact of Cl, while

the reduction itself is better to quantify the model ability to match the observations. We compute the reduction ratio using the following equation :

$$r = \frac{(m_{\text{REF}} - m_{\text{CHL}})}{(m_{\text{REF}} - o)} \times 100 \qquad (6)$$

$r$ denotes the reduction ratio (or relative difference), $m_{\text{CHL}}$ is the simulated value from the TOT_CHL simulation, $m_{\text{REF}}$ is the simulated value from the TOT_REF simulation and $o$ is the observed value. The smaller the value of $r$, the better is

the match between model results and observations. We compute the difference between simulated and observed profiles after the simulated values have been linearly interpolated on the AirCore profile pressure axis. Points outside the data range are extrapolated using the inferred interpolation function. The model-observation discrepancy distributions are shifted to the left

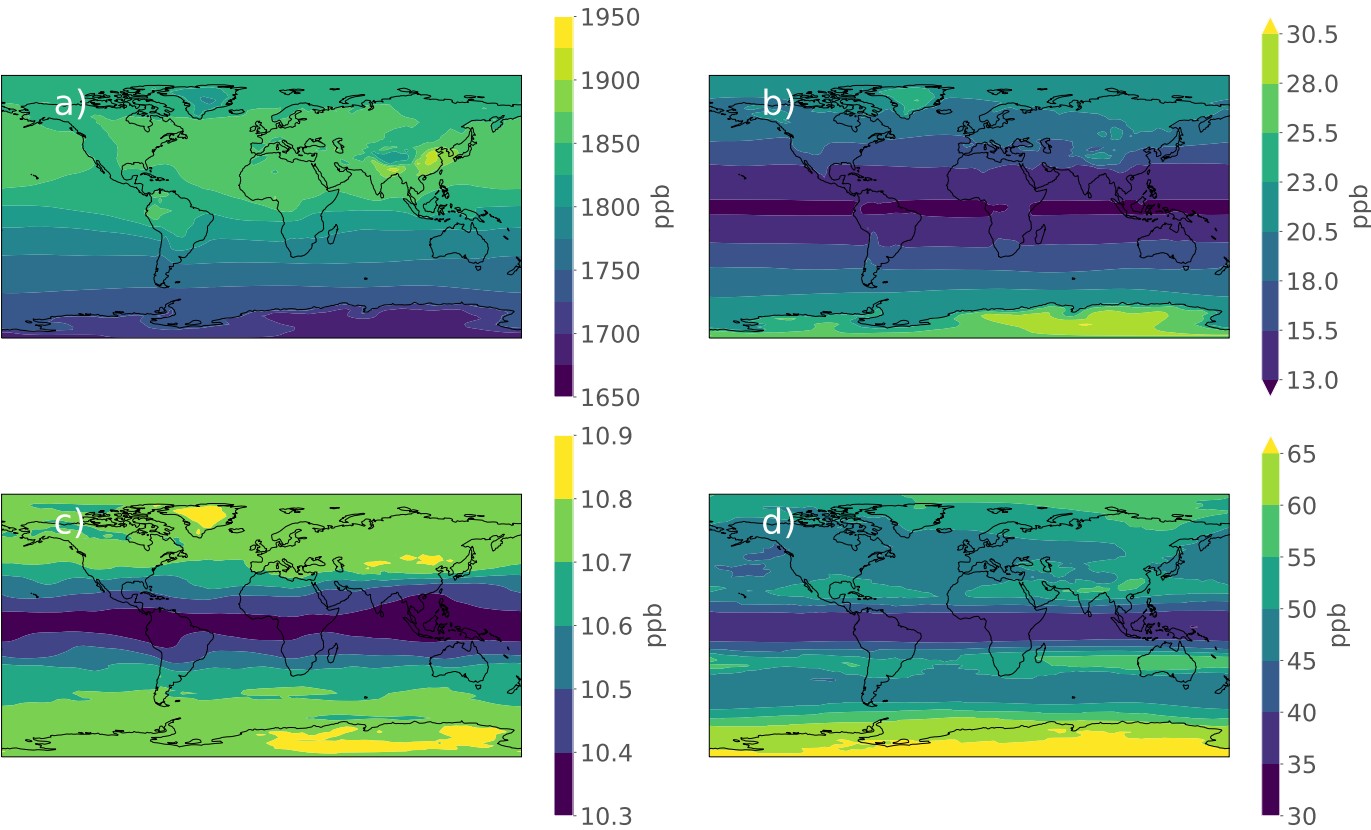

**Figure 4.** Dry-air column average mole fraction of methane (a) and differences of methane total and partial columns between simulations (TOT_REF - TOT_CHL) averaged over 2018 in ppb ; a) Dry-air column average mole fraction of methane TOT_CHL. b) Dry-air column average mole fraction difference. c) Tropospheric methane partial column difference. d) Stratospheric methane partial column difference.

towards smaller values (reduction ratio smaller than 100%) when including the Cl sink for all seasons in both the troposphere and stratosphere (Fig. 5). This illustrates the improvement of the model when including Cl. We find that the implementation of the Cl sink reduces the difference between simulated and observed vertical profiles on average by 26.7 ± 9.73 % (1σ for intra-profile variability) in the stratosphere over the entire dataset and by 38.3 ± 13.8 % in the troposphere. This ratio represents

5     a $CH_4$ reduction of about 24 ppb in the stratosphere and 8.1 ppb in the troposphere. However, even including the Cl sink, the simulations shows significant remaining discrepancies in the troposphere compared to AirCore observations (∼20 ppb, 1,1% of the tropospheric mixing ratio) and in the stratosphere (∼85 ppb, 5.5% of the stratospheric mass-weighted mean mixing ratio).

    The reason of this systematic error in the troposphere might be due to the overestimation of the emission scenario from

10     GCP which reflects the state of the art of emission scenarios but is not optimized against atmospheric observations and not in adequate balance relative to the chemical sinks in the model. Since 1) the errors are much larger in the stratosphere than in





the troposphere and 2) the correction of the tropospheric bias would only (very likely) slightly shift the stratospheric values to smaller values, not correcting for this systematic error will not affect the conclusions in this work. Performing similar comparison using optimized fluxes based on inversions including the Cl sink would help to overcome this issue. However the main objective of our study is to evaluate the impact of the Cl sink and not really to fit to the observations. Further work will be

dedicated to better match the observations based on the results of atmospheric inversions from Saunois et al. (2019) and new ones using isotopic constraint. The underestimated Cl concentration in the MBL and more generally in the troposphere could also contribute to the bias simulated in the lower troposphere. Improving our estimates of Cl concentrations in the troposphere may help reducing this tropospheric bias, which spreads into the stratosphere and is amplified by the discrepancies between observed and simulated vertical gradients.

Figure 5 shows the seasonal distribution of the differences (and reduction ratio) between the simulated and observed by AirCore methane mixing ratio for the tropospheric and stratospheric layers with and without the Cl sink. The AirCore datasets provide multiple profiles for each season. Seasons are referred hereafter as North Hemispheric seasons, i.e winter meaning boreal winter. Note that an AirCore profile retrieved above South Hemisphere locations is included in the list associated to its true season. For instance, an AirCore dataset retrieved during January in the South Hemisphere is added to the summer list.

The displayed value in each frame is the mean value of these differences over the considered region, namely troposphere or stratosphere. The reduction ratio (third line and sixth line) ranges from 32 % (spring) to 43% (autumn) in the troposphere and from 23% (spring) to 33% (winter) in the stratosphere. Therefore, we have larger reduction ratios in the troposphere than in the stratosphere, regardless of the season. The reduction is in a range of 6.69 (autumn) - 10.50 (spring) ppb in the troposphere and 19.26 (winter) - 28.04 (summer) ppb in the stratosphere. The reduction is thus much lower (more than 10 ppb) in the

troposphere than in the stratosphere. Nevertheless, the reduction in each region of the atmosphere is dependent on the season. Moreover, spring always exhibits the largest discrepancies after the Cl sink has been implemented. Latitudinal dependence have not been analyzed since the numbers of AirCore profiles retrieved in high-latitudes regions is very low (see Figure 2).

   Figure 6 displays the comparison between AirCore observations and associated simulated profiles (with the Cl sink implemented) linearly interpolated on a regular altitude axis following the same method as before. Above the tropopause between

15-25 km, the difference can exceed 150 ppb on average during spring and summer even with the Cl implemented. All the profiles and their associated simulated profiles are plotted in Figure S3. A closer analysis of individual profiles shows that more small-scale variations occur during spring and summer than during other seasons, especially around the tropopause. Additionally, inversions of concentrations can occur above the tropopause. Those may be induced by small scale filaments of methane coming from polar regions, that cannot be adequately reproduced by our low-resolution model and seasonal averaging.

The profiles of vertical gradient are plotted in Figure 6c. The values at the top of the profile (between 25 and 30 km) have been removed. Indeed, those values cannot be easily interpreted due to the AirCore sampling methodology. We see that the simulated vertical gradient turns positive higher in altitude in comparison with the observations (∼ 16-18 km for simulated gradients against 13-16 km for observed gradients), except for the simulated gradients in the DJF period, which are very close to observed gradients until 20 km. This altitude difference leads to a positive difference between the observed and the simulated gradient. The simulated gradient tends to readjust about 3-4 km higher in altitude (null difference) and then the gradient






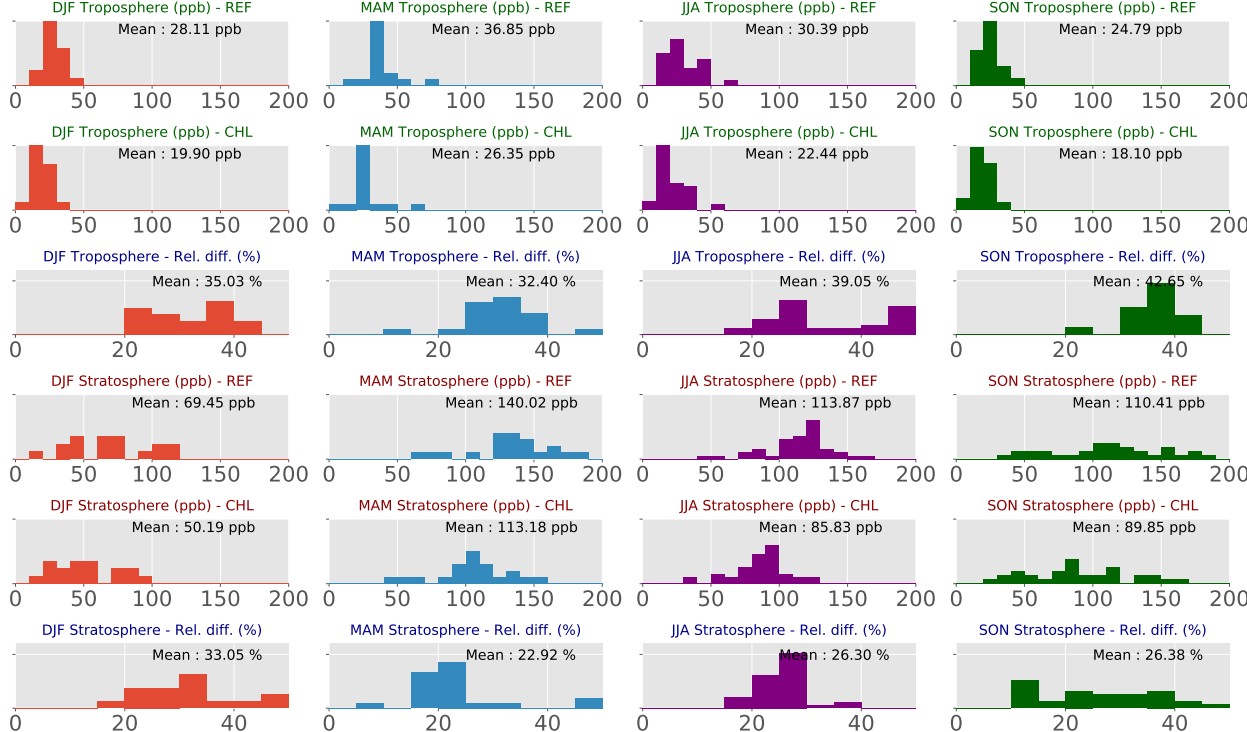

**Figure 5.** Distribution of the difference model-obs over the AirCore profiles. For each profile, we interpolate the model vertical profiles on the pressure grid of the profile and calculate the difference of mixing ratios. We then compute the mean of the differences for a specific region of the atmosphere. The three first lines shows the differences in the troposphere and the last three ones the impact in the stratosphere. The first line of each group shows the distribution of the absolute model-obs difference in the troposphere for the REF simulations for each season (without the chlorine sink). The second line shows the same for the CHL simulations (with the chlorine sink). The third line shows the reduction ratio according to the equation (6). The three last lines show the same features as the three first lines but in the stratosphere. The mean over the dataset is reported in each frame.

difference seem to turn negative above 20 km, meaning the simulated gradients might be overestimated at this level. Patra et al. (2011) and Thompson et al. (2014) pointed out that the LMDz version with 19 levels exhibited a too fast Stratosphere-Troposphere Exchange (STE) and a wrong tropopause height compared to other CTMs. (Locatelli et al., 2015) showed that increasing the number of levels from 19 to 39 levels acted to improve the flaw of LMDz regarding STE and tropopause height. However, the analysis of this large dataset of vertical profiles retrieved during different seasons suggests that STE errors are still there. Furthermore, the discrepancies between simulated and observed mixing ratios at an altitude of 20 km can exceed 200 ppb, likely due to an underestimation of the Brewer-Dobson circulation intensity that transport the methane upward into the upper stratosphere. The overestimation of the vertical gradient above 20 km also supports this hypothesis. Switching to





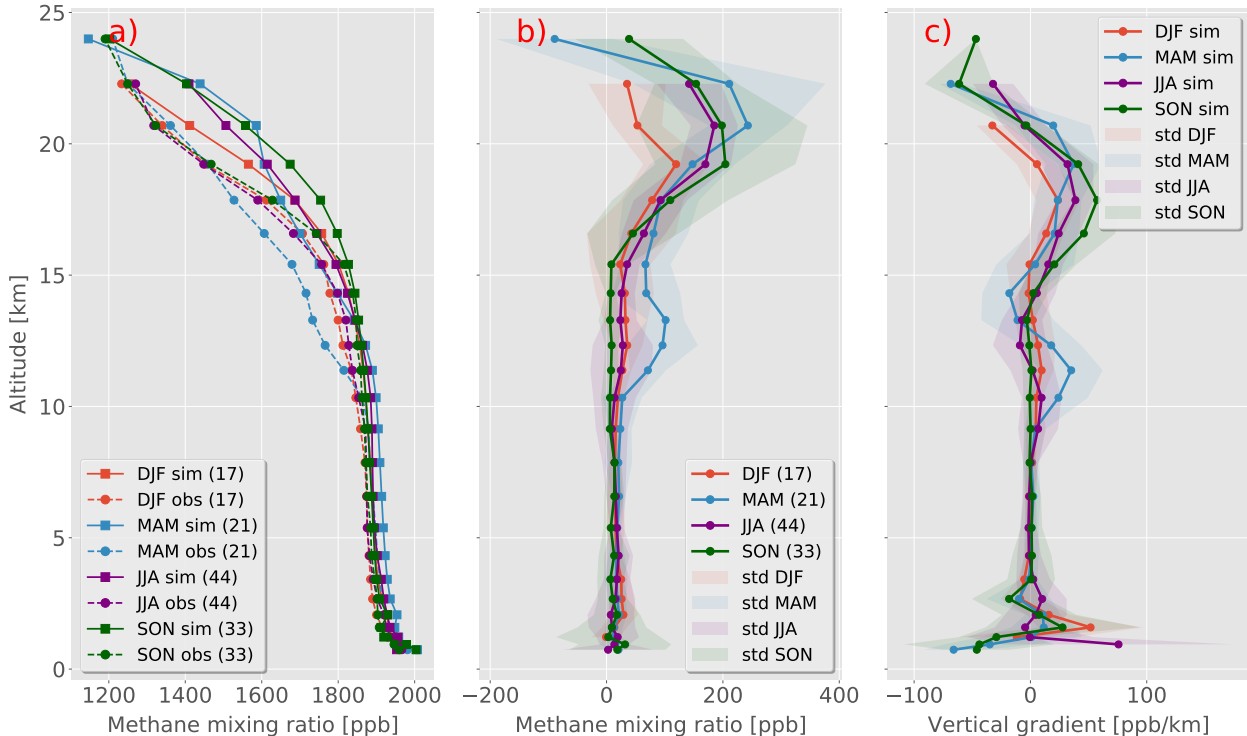

**Figure 6.** a) Mean simulated and observed profiles in ppb. The figure in brackets gives the number of Aircore profile retrieved for each season. b) Absolute difference between simulated and observed profiles in ppb. c) Difference between vertical simulated and observed gradients (sim - obs) in ppb/km. All the profiles (simulated and observed) has been averaged over each season.

isentropic coordinates as suggested by Patra et al. (2011) is more likely to reduce this error than increasing the number of vertical levels.

## 3.4 Impact of Cl on stratospheric $\delta^{13}$C-CH$_4$

The present offline version of LMDz should be able to properly simulate $^{13}$C:$^{12}$C ratio if future inversions are to be run with
5    $\delta^{13}$C-CH$_4$ as a supplementary constraint. The simulated isotopic signal depends strongly on the estimated sinks, of the isotopic signature chosen for the sources, the relative magnitude of the sources and on the atmospheric transport. Among the sinks, the Cl sink is the most strongly fractionating one (see Table 1). Therefore, even though the impact of Cl on total methane is small in comparison to OH, Cl is expected to greatly influence the $^{13}$C:$^{12}$C ratio, especially in regions where the Cl sink is predominant. Here we compare our simulated results to the observations from Röckmann et al. (2011) (Table 4). The results of
10  the comparison are shown on Figure 7. Implementing the Cl sink drastically improves the simulation-observation comparison. The fact that $\delta^{13}$C-CH$_4$ stratospheric vertical profiles improve when implementing the Cl sink was already demonstrated by McCarthy (2003). However, it shows that LMDz models the processes affecting $\delta^{13}$C-CH$_4$ overall in a realistic manner.





Nevertheless, Figure 7 suggests that there may be a remaining underestimation of the 13C vertical gradient in the lower stratosphere and an overestimation above. The poor representation of the Brewer-Dobson circulation highlighted in Section 3.3 could explain this issue. Indeed, if too much methane is trapped at the tropopause level, the ratio $^{13}$C:$^{12}$C will be reduced, and then underestimated. As a result, less methane is simulated above the tropopause, leading to an overestimation of the $^{13}$C:$^{12}$C ratio. Other possible explanations would be that the values taken for the KIE are not reliable enough at these altitudes or that Cl concentrations are poorly estimated at these levels.

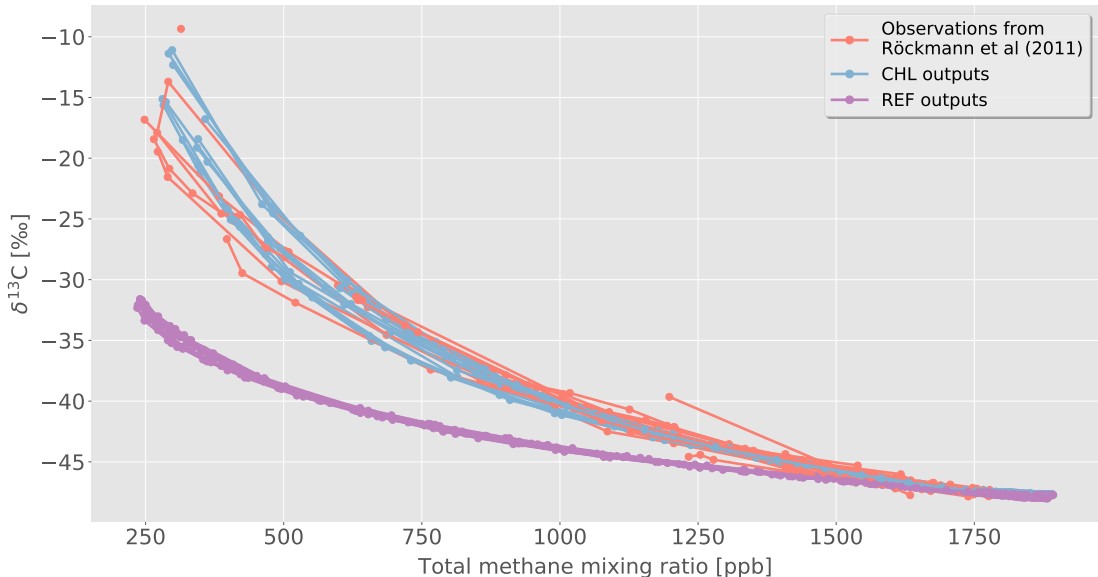

**Figure 7.** Global mean $\delta^{13}$C with respect to total methane mixing ratio. Comparison of $\delta^{13}$C simulations outputs with observations from Röckmann et al. (2011)

### 3.5 Impact of Cl on surface $\delta^{13}$C-CH$_4$

To assess the impact of Cl on global surface values of $\delta^{13}$C-CH$_4$, we run a set of six sensitivity tests (Table 5) over the 2000-2018 period and show the time series of the surface global mean $\delta^{13}$C-CH$_4$ signal in Figure 8.

Note that the initial Cl field used in the d13_CHL simulation exhibits a global tropospheric mean value of 240 atoms.cm$^{-3}$ and a MBL mean value of 130 atoms.cm$^{-3}$. In the S5 scenario, the tropospheric Cl is removed. In S6, the stratospheric Cl is removed. In S7, the tropospheric global mean of Cl concentrations is scaled to 620 atoms.cm$^{-3}$, as reported byWang et al. (2019). In S8, the tropospheric mean is scaled to the values of Wang et al. (2019), namely 620 atoms.cm$^{-3}$ in the troposphere and 1200 atoms.cm$^{-3}$ in the MBL. In S9, the Cl concentrations are decreasing by 5%/decade from 2000 to 2018. Finally, S10





is the same as the latter but with the tropospheric concentrations from Wang et al. (2019). We apply the same emissions for the entire period, so the differences between the final values of each scenario are only correlated with the Cl concentrations.

### 3.5.1 Impact of tropospheric chlorine on surface $\delta^{13}$C

As expected, the global surface value of $\delta^{13}$C-CH$_4$ is found to be positively correlated with the Cl tropospheric mean concentration. d13_REF, d13_CHL, and S7 exhibit final values (trend) in 2018 of -47.34 ‰ -46.98 ‰ -46.78 ‰ respectively. Results of S8 show to what extent the MBL Cl concentrations influence the surface $\delta^{13}$C-CH$_4$. Using a simple linear relationship between MBL concentrations and surface $\delta^{13}$C-CH$_4$, we could expect a final value for S8 of -45.94 ‰. Instead, we obtain a final value of -46.58 ‰ showing that the tropospheric mixing acts to reduce the impact of MBL Cl on surface $\delta^{13}$C-CH$_4$. Despite this reduction, the tropospheric Cl sink shows a great influence on surface $\delta^{13}$C-CH$_4$ and should be considered. The differences between these sensitivity simulations are much larger than observed changes in $\delta^{13}$C-CH$_4$ global mean (Fig. 8). Indeed, the observed globally-averaged $\delta^{13}$C-CH$_4$ ranges between -47,04 ‰ and -47,38 ‰ during the 2000-2018 period (thick blue line in Fig. 8). Thus the difference between d13_REF and S8 is equal to 250 % of the $\delta^{13}$C-CH$_4$ min-max observed range. Therefore, not considering surface Cl in an inversion can potentially lead to a significant underestimation of the weight of biogenic sources or an overestimation of the weight of anthropogenic sources (apart from livestock sectors) in the global budget since the Cl sink tends to enhance $\delta^{13}$C-CH$_4$ values. Simulating the same scenarios using the wetland signature map provided by Ganesan et al. (2018) (a mean value of -60.8 ‰ using our fluxes) shows no significant change in the differences between the final values (not shown here) besides shifting all the values towards more negative ones (difference of -0.84 ‰ for all the final values).

With our wetlands isotopic signature map, the scenario giving the best agreement with observations is S6 (only tropospheric Cl) even if the post-2007 negative trend is not reproduced. With wetlands isotopic signature from Ganesan et al. (2018) (simulations not shown here), the scenario giving the best agreement with observations is S8 (Cl concentration values from Wang et al. (2019)).

### 3.5.2 Impact of stratospheric Cl on surface $\delta^{13}$C-CH$_4$

The circulation in the upper troposphere and the lower stratosphere (UTLS) can be, to first order, described by an upwelling from the troposphere to the stratosphere in the tropics, a meridional circulation in the stratosphere to the extratropics and a downwelling from the stratosphere to the troposphere in middle and high-latitudes. (Bönisch et al., 2011; Stohl, 2003). As methane is transported upward and throughout the stratosphere, it becomes heavier (less negative $\delta^{13}$C-CH$_4$ values) because of fractionation of atmospheric sinks. Then, as air is injected back to the troposphere, this heavier air and the tropospheric light air start mixing. Therefore, in addition to the enhanced $\delta^{13}$C-CH$_4$ stratospheric signal, the stratospheric Cl should also have a significant impact on the tropospheric $\delta^{13}$C-CH$_4$ values.

The difference between the simulations d13_REF (no Cl at all) and S5 (no Cl in the troposphere) exhibits the impact of stratospheric Cl through stratosphere to troposphere air injections. At the end of the time series, the deseasonalized trends have a difference of 0.27 ‰ which represents 80% of the min-max $\delta^{13}$C-CH$_4$ observed range. Hence, the stratospheric Cl impact





on the surface $\delta^{13}$C-CH$_4$ (thus lower than the previously estimated value of 0.5 ‰ by McCarthy et al. (2001)) is significant and running an inversion using an isotopic constraint and without implementing a realistic stratospheric Cl sink could result in, for instance, an underestimation of biogenic sources intensities.

One possible assumption would be that the decrease of the $\delta^{13}$C-CH$_4$ since the year 2007 across the globe is not completely
due to the increase of biogenic sources (Nisbet et al., 2016) but could be partly attributed to the decrease of stratospheric reactive Cl since the Montreal Protocol. Indeed, reduced stratospheric Cl would lead to lower $\delta^{13}$C-CH$_4$ surface values. Bernath and Fernando (2018) have analyzed recent observations of the stratospheric HCl mixing ratio and concluded that it decreases by about 5%/decade. Even though HCl (or total Cl) and atomic Cl are not so simply correlated, applying the same decrease to our Cl field could give us some insight into the potential impact of a recent decrease in stratospheric Cl on $\delta^{13}$C-CH$_4$ values at the
surface. The scenario S9 (low tropospheric Cl and decreasing stratospheric Cl) shows a reduction of the surface $\delta^{13}$C-CH$_4$ value of only 0.02 ‰ compared to the scenario d13_CHL and the scenario S10 (high tropospheric Cl and decreasing stratospheric Cl) shows a reduction of 0.03 ‰ compared to S8. Hence, this shift resulting from the Montreal Protocol ratification is not likely to be a cause of the recent decrease of $\delta^{13}$C-CH$_4$ values towards more negative values (about 0.3 ‰ over the last 10 years, Nisbet et al., 2019).

All things considered, if $\delta^{13}$C-CH$_4$ constraint are to be used for sources characterization in long-period inversion runs, the simulated stratospheric impact of 0.27 ‰ when including stratospheric Cl sink should be considered.

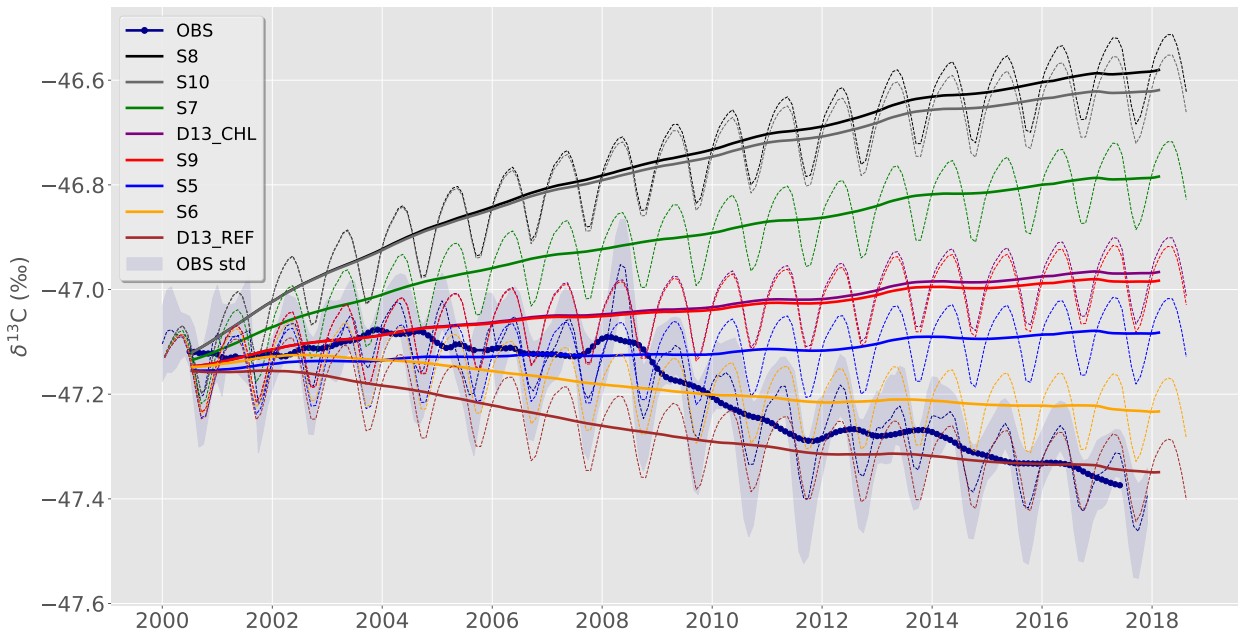

**Figure 8.** Time series of surface $\delta^{13}$C global mean value for multiple scenarios. Dashed lines are monthly values and solid lines are deseasonalized trends. The globally-averaged NOAA-GGGRN $\delta^{13}$C record is in dark blue. The shaded area represent the standard deviation of the observed records.





## 4 Conclusions

This study presents the impact of atomic Cl on the modeling of total methane removal and on $\delta^{13}$C-CH$_4$ using LMDz. Three methane observational datasets have been used to assess this impact : 115 AirCore vertical profiles, $\delta^{13}$C-CH$_4$ stratospheric measurements from Röckmann et al. (2011) and $\delta^{13}$C-CH$_4$ 2000-2018 records from 18 ground stations distributed all over

the world. 10 forward simulations (including 6 sensitivity test runs) have been assessing the impact of Cl field on methane simulated concentrations. The Cl stratospheric sink strength is estimated to be 7.16 ± 0.27 Tg/yr, accounting for 29% of the total stratospheric sink. Implementing the Cl sink has also an effect on methane total column (XCH$_4$) of about 1% (18 ppb) after a 13-year simulation period (2006-2018). Even though the Cl sink reduces the discrepancies between AirCore and simulated methane vertical profiles, large discrepancies in both tropospheric (likely mostly due to the non-optimized emission

scenario) and stratospheric CH$_4$ still remain. LMDz has difficulties to reproduce small-scale variations exhibited during boreal spring and summer. Moreover, the Brewer-Dobson circulation that governs lower to upper stratosphere air transport is only fairly reproduced. We also showed that the isotopic ratio $^{13}$C:$^{12}$C ratio is more substantially affected by the Cl than total CH$_4$. The stratospheric vertical profiles of $\delta^{13}$C-CH$_4$ values agree very well with observations from Röckmann et al. (2011) when including the Cl sink. At the surface, the set of the sensitivity tests performed with or without the stratospheric or tropospheric

Cl sink show that the Cl concentrations at the surface can largely affect the $\delta^{13}$C-CH$_4$ surface signal. Indeed, there is a difference of 0.76 ‰ between a Cl-free simulation and a simulation with Cl values from Wang et al. (2019) (last values of each detrended time series). This influence also raises the question of the uncertainties on the source isotopic signatures. Considering a 1-D model, the final $\delta^{13}$C-CH$_4$ surface value ($\delta^{13}C_{\text{surf}}$) can be easily inferred from the global KIE$_{\text{app}}$ and the mean (flux- and -area weighted) signature $\delta^{13}C_{\text{source}}$ using the equation below :

$$\delta^{13}C_{\text{surf}} = (1 + \delta^{13}C_{\text{source}}) \times \text{KIE}_{\text{app}} - 1 \tag{7}$$

A $\delta^{13}$C-CH$_4$ source delta (difference) could produce the same effect as the Cl-induced shift of 0.76 ‰. This $\delta^{13}C_{\text{source}}$ change could be caused by a decrease in the biogenic sources intensities and/or an increase in these sources isotopic signatures. Taking the example of wetlands in our study, decreasing the share of the wetlands in the total budget of 7% or increasing the isotopic signature of +2 ‰ would lead to a 0.76 ‰ $\delta^{13}C_{\text{source}}$ increase. Hence, implementing a realistic Cl field in a poorly

signature-constrained inversion would have limited impacts on the final result uncertainty. Unfortunately, wetland isotopic signatures vary widely from one study to another at the global scale, going from -60.5 ‰ Feinberg et al. (2018) to -62 ‰ Ganesan et al. (2018) and we only discuss here the impact of global average values, as local to regional signatures can vary over much larger ranges. A box model would be too simple to rigorously study the impact of regional uncertainties. In addition, the stratospheric impact of Cl on surface $\delta^{13}$C-CH$_4$ values is as high as 0.27 ‰ of the same magnitude as recorded variations

of $\delta^{13}$C-CH$_4$ in the past decade. However, recent Cl concentration decrease owing to the Montreal Protocol are not likely an explanation for the recent shift of $\delta^{13}$C-CH$_4$ values towards more negative values.

We should also not ignore the errors in the estimation of total methane in UTLS. This problem has been resolved by some modellers by implementing isentropic coordinates, best suited for Brewer-Dobson circulation representation, but are unlikely

to be done in LMDz in the near future due to technical considerations and a way to virtually move methane upwards into the upper stratosphere should be rather considered. Future work will focus on running inversions with isotopic constraints to better characterize the various methane sources and sinks, with the use of spatially resolved isotopic signature maps as suggested by Feinberg et al. (2018) and Ganesan et al. (2018) in order to limit the errors generated by a poor representation of these

elements. Some sensitivity tests using different OH and Cl fields should be also considered in order to quantify the impact of sinks uncertainties on inversion results.

*Data availability.* The data for $\delta^{13}$C-CH$_4$ observations were downloaded from the World Data Centre for Greenhouse Gases (WDCGG) at https://gaw.kishou.go.jp. Datasets for the input emissions were provided by the Global Carbon Project (GCP) team. The AirCore vertical pro-files from the NOAA-ESRL Aircraft Program (v20181101) were provided by Colm Sweeney and Bianca Baier. The Cl fields, the modelling

output files and the AirCore vertical profiles from the French AirCore Program are available upon request from the corresponding author.

*Author contributions.* JT designed and run the simulation experiments and performed the data analysis presented in this paper. DH provided the Cl fields used for the simulations. The AirCore data was retrieved and provided by MR and CR (from the French AirCore Team) and CS and BB (from the NOAA-ESRL Aircraft Program, v20181101). MS provided the total methane fluxes. MS, AB, IP and PB provided scientific and technical expertise. They also contributed to the scientific analysis of this work. JT prepared the manuscript with contributions

from all co-authors.

*Competing interests.* The authors declare that they have no conflict of interest.

*Acknowledgements.* This work was supported by the CEA (Commissariat à l'Energie Atomique et aux Energies Alternatives). We are grateful to Thomas Röckmann for sharing his data and for insightful discussion. The study extensively relies on the meteorological data provided by the ECMWF. Calculations were performed using the computing resources of LSCE, maintained by François Marabelle and the

LSCE IT team. The authors wish to thank the measurement teams from the Global Greenhouse Gas Reference Network (NOAA) for their work.



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
