# Peer review of "Impact of atomic chlorine on the modelling of total methane and its 13C:12C isotopic ratio at global scale"

_Atmospheric Chemistry and Physics, 2019_

## Referee Comment (RC1) · Anonymous Referee #2 · 5 Dec 2019

In their study, Thanwerdas et al. attempt to gain a deeper insight on the importance of atmospheric Cl for CH4 mixing and isotope ratios distribution from the surface up to upper stratosphere/mesosphere, using a comprehensive 3D GCM with parameterisation of CH4 sources and sinks. The topic and research question here are certainly within the scope and interest for ACP, and I find the sensitivity experiments proposed conceptually useful. However, I cannot judge the adequacy of obtained results in this study because model setup/spin-up raises questions (I hope mostly owing to merely poor quality of the presentation), see the comments below. The devil is in the details, and the latter are not clear.

[Figure]

The reviewer will certainly reconsider this manuscript again after 1) a substantial improvement of the clarity of scientific statements used (most of specific comments below address these), 2) resolution of the questions w.r.t. model setup (general comments). It is necessary to let a native English speaker edit and improve the entire manuscript; in the current state, it is difficult to read (excessive use of definite articles, clumsy sentence composition, etc.). Please understand that the reviewer has limited time for reviewing, therefore the next iteration will have more focus on the results.

General comments

P3L29: I am concerned about the adequacy of the model setup, namely the adopted INCA fields: are these obtained using consistent atmospheric dynamics (i.e. driven from similar nudging data)? If not, oxidants fields may quite not correspond the offline model.

P7L12-17: if there is an explicit representation of CH4 isotopologues in the model, why do you use "negative emission" apparatus and Eq.(5) instead of explicitly simulating soil uptake via isotopologues with respective fractionation in the surface layer? I cannot confirm the correctness of the approach used here, because I do not see what for $\delta$13Ceff is introduced. From Eq.(5) it follows that $\delta$13Ceff is the isotope composition of the uptake flux, depleted in 13C w.r.t. to the surface layer burden. The latter becomes enriched in 13C upon uptake, in proportion to the removal flux. The dynamic equilibration of these processes results in $\delta$13Camb, however you cannot take the latter in Eq.(5) because you will introduce additional fractionation. In other words, the uptake flux should be calculated from 13C/12C simulated in the model, not the "ambient" signature. If you are certain this is an admissible approach, please quantify the error introduced it.

P11L1-3: Arriving at a proper dynamically equilibrated isotope CH4 distribution up to mesosphere for a given year of the present is a difficult task (also because of not equilibrated CH4 burden). History of CH4 transport is still present in the UTLS (where

[Figure]

CH4 lifetime can reach 100 years) up to middle stratosphere, so taking just year 2000 emissions is not a satisfactory approach (as compared to, e.g., at least taking 1980-2000 CH4 emissions). A least uncertain way here is to prescribe transient mixing and isotope ratios at the surface and let the model run for 1980-2000 to "populate" atmosphere with CH4 isotopologues, in hope that transport and atmospheric sinks are adequate. To recap: to ascertain the results of simulations starting from such obtained initial conditions, you need to show in this work that your spin-up method is adequate, i.e. by estimating errors or comparing to other simulations using pre-2000 CH4 emissions and atmospheric distribution.

A more concerning issue is the subsequent "adjustment of the simulated isotopic signal" to "obtain satisfying initial conditions". How is this adjustment performed and on which grounds? I am not aware of a method that allows adjusting 3D fields of 12CH4 and 13CH4 to some surface observations. The non-linearity of transport, mixing and fractionating sink processes does not allow this. Furthermore, if such adjustment is required, it implies your initial conditions for the experiments are erroneous. Until this issue is clarified, I see no point considering the results of the simulations.

P113-4: I do not understand why constant emissions for 2000 are used? In Sect. 2.4 you note that transient emissions since 2000 are prepared. You also report figures (although averaged) for 2006-2018.

Specific comments

P1L6: for a non-specialist, "very fractionating" will be very unclear

P1L8: you probably mean "General Circulation Model (GCM)"

P1L9: there is ONE major sink of CH4 in the atmosphere, it is reaction with OH; I suggest using "all relevant CH4 chemical sinks" instead of "major . . . reactions"

P1L10: it is better to add year for which sink estimate is given

P1L15: less negative → enriched in 13C

P1L21: "globally averaged concentrations" → atmospheric concentrations/burden

P2L4-5: do you mean "resumed increase"?

P2L13: you mention mostly inverse modelling studies, however here you use forward modelling; I believe it is important to review/refer to the forward modelling studies using CH4 isotopes, too (I can now remember only NIWA group that is missing here, e.g. Schaefer et al. (2016, doi:10.1126/science.aad2705), though there are more)

P2L14: I would not use "exactly" here, their conclusions are substantially different!

P2L17-18: "light signature" is not easy to comprehend for non-isotope specialist, reformulate; regional variability in isotope composition only, or also in emission strengths?

P2L18: define or reformulate "fractionation potential"

P2L20: not entirely correct statement, CH4 sinks also via O(3P) and, more importantly, in photolysis reactions in upper atmosphere; better specify domain for clarity

P2L30: impact of Cl on what?

P2L31: are there no 3D model focussing on Cl impact in the stratosphere to date???

P3L4: "total methane" is vague, use "abundance/mixing ratio"

P3L14: are winds nudged in the online version and then fed to the offline version of the model?

P3L22: what is meant by "each reaction is bound to a reaction type"?

P3L24-26: please specify which species are prescribed, this is important

P3L31: "its isotopologues" comprise CH4, use "its isotope ratio"

P4L1: reformulate the section caption, e.g. "Kinetic Isotope Effect" → "isotope fractionation"; isotope effects are different in each reaction due to different reaction mechanisms

P4L7-8: you seem to convey the message that reaction of CH4 with Cl IS important for the tropospheric composition? Please use "mixing ratio" instead of "total methane"

P4L2: see comment to P2L20

P4L9: "can vary" → "varies" (we have lab studies confirming that)

P4L18-19: it is not clear what the "effective isotope signature" is and how it is used (see comment to P7L12-17)

P5L31-32: please, reformulate and expand this statement (also do not use "should"). Do you intend so say that the large variations shown in Fig. 1 are expected to play a key role in simulated CH4 seasonality? If so, I do not see why – the other oxidants are also strongly photochemically driven. Important: please explicate which mechanism lead to increased photolysis within polar vortices?

P6L10: what is the sense of showing averages for this period? ranges would be much more useful

P6L13-14: "global budget" usually implies atmospheric burden split by emission type; I suggest using "total emission"

P7L1: signatures of what?

P7L15: define "isotopic signal" or you mean isotope composition?

P7L19-20: it is important to mention whether the signatures of compiled emissions exhibit any significant temporal trends; if there are any, perhaps they could be illustrated next to Figure S2?

P7L20: unclear sentence, you set the same (i.e. one) constant signature for all other emission categories?

P8L16-18: on which grounds you make this assumption? e.g. conditions within and outside polar vortex can vary substantially, which one would have to take into account

for high-latitude stations

P8L13-: please quote uncertainties associated with measured 13C/12C ratios, too

P10L13: is the "total" CH4 set equal to sum of 13CH4 and 12CH4, or the latter are scaled up at the same ratio? This question is important for upper stratospheric budgets, where fractionation is high and mixing ratios are low…

P10L14: adjust to what? Be clear about your statements! Note, Tans (1997) paper is about hemispheric equilibration of abundances and isotope ratios in CH4 in idealised model runs.

P11L5: what is the difference between the pairs of TOT_ and d13_ simulations? Because you state that total CH4 is a sum of 13CH4 and 12CH4 in your setup, why one has to simulate TOT_ ?

P11L8-13: these two simulations should have different results; which are reported in this paragraph? Furthermore, what is the sense of making a simulation without Cl, when its contribution can be simply inferred from the model? Removing Cl may shift the photochemistry of the model to much more unrealistic state, you have to show that this is not the case. I see more sense in a sensitivity simulation where CH4+Cl reaction does not have 13C fractionation.

P12L4-5: it is not concluded in Sect. 2.3 (P5L27-30) that the tropospheric Cl in your runs is underestimated; it is lower than in two of three studies referenced

Technical comments

P2L11: wordiness, "can potentially allow"

P2L15: too much of definite articles

P2L25: use "13CH4:12CH4 ratio"

P2L26: "on methane mixing and isotope ratios"

P3L30: remove "here"

P4L17: unclear sentence, which "those values" are implied?

P4L18: chemical removal (singular use)

P5L10: perhaps, no new paragraph?

P5L12: "dependent"

P6L2-5: sentence is difficult to read, "emissions" and "prescribe surface fluxes" are essentially the same

P6L12: too busy sentence, use "considered time" or add a comma

P7L18: "sectors" usually refer to sub-categories of anthropogenic sources; please, use "emission category" or similar but not "sectors" (these are also not mentioned in Table 2) and check for unambiguous terminology throughout the manuscript

P7L17 new paragraph?

P7L24: remove "can"

P10L1: "to provide isotope measurements" → "for CH4 isotope composition"

P10L1-3: totally unclear sentence, grammar? how much is "enough"?

P10L8: parenthesise simulation abbreviations

P12: tropopause is shown with red line (green in caption)

//

––––––––––––––––––––––––––––––

---

## Referee Comment (RC2) · Anonymous Referee #3 · 4 Feb 2020

Please find review comments on the Supplement.

Please also note the supplement to this comment:
https://www.atmos-chem-phys-discuss.net/acp-2019-925/acp-2019-925-RC2-supplement.pdf

---

## Author Comment (AC1) · 16 Mar 2020

We thank the referees for their time and for giving fruitful comments and reviews to our manuscript. It helped improving our manuscript substantially. We address in the attached file their comments and our corresponding corrections to the manuscript. Comments from Referees #1 and #2 are reported below in blue and red respectively. We include point-by-point replies and corresponding corrections to the manuscript are included in bold between horizontal lines. The marked-up manuscript is added at the end of the attached pdf.

[Figure]

Please also note the supplement to this comment:
https://www.atmos-chem-phys-discuss.net/acp-2019-925/acp-2019-925-AC1-supplement.pdf